# Comparative Evaluation of the Efficacy of Two Ectoparasiticides in Preventing the Acquisition of *Borrelia burgdorferi* by *Ixodes scapularis* and *Ixodes ricinus*: A Canine Ex Vivo Model

**DOI:** 10.3390/microorganisms12010202

**Published:** 2024-01-18

**Authors:** Djamel Tahir, Virginie Geolier, Sophie Dupuis, Nouha Lekouch, Elisabeth Ferquel, Valérie Choumet, Marie Varloud

**Affiliations:** 1Environnement et Risques Infectieux, Institut Pasteur, Université Paris Cité, 75015 Paris, France; virginie.geolier@pasteur.fr (V.G.); elisabeth.ferquel@pasteur.fr (E.F.); valerie.choumet@pasteur.fr (V.C.); 2Ceva Santé Animale, 10 Avenue de la Ballastière, 33500 Libourne, France; sophie.dupuis@ceva.com; 3Clinvet Morocco, B.P 301, Mohammedia 28815, Morocco; nouha.lekouch@clinvet.com

**Keywords:** Lyme borreliosis, dog, *Ixodes scapularis*, *Ixodes ricinus*, ectoparasiticide, vector control, tick-borne disease prevention, ex vivo model

## Abstract

In dogs, tick infestation can cause damage ranging from a simple skin irritation to severe diseases and/or paralysis leading to animal death. For example, *Ixodes ricinus* and *I. scapularis* are among the tick species incriminated the most in the transmission of *Borrelia burgdorferi*, the agent of human and canine Lyme borreliosis (LB). In this study, we aimed to compare the efficacy of two products designed for dogs—an oral systemic ectoparasiticide and a topical repellent ectoparasiticide—against the acquisition of *B. burgdorferi* by adult *I. scapularis* and *I. ricinus* using an ex vivo model. Thirty-two beagle dogs were included in a parallel-group-designed, randomized, single-center, negative-controlled efficacy study. The dogs were allocated to three groups based on gender and body weight: a fluralaner (F, Bravecto^®^) treatment group (*n* = 8), administered a single oral treatment on day 0 at the recommended dose; a dinotefuran–permethrin–pyriproxyfen (DPP, Vectra^®^ 3D) treatment group (*n* = 8), topically treated on day 56 at the recommended dose; and an untreated control group (*n* = 16). Blood and hair were collected from each dog on days 58, 63, 70, 77, and 84. Hair was added to the silicone-based membrane separating two glass chambers forming the feeding unit (FU). Chamber 1 was filled with blood spiked with *B. burgdorferi* sensu stricto, strain B31 (10^5^ cells/mL). Chamber 2, glued below chamber 1, was seeded with 20 adult *I. scapularis* or *I. ricinus*. The FUs (*n* = 240) were incubated at 37 °C with a humidity >90%. Tick survival, attachment, and feces presence were observed from 1 h up to 72 h after tick seeding. The uptake of *B. burgdorferi* was determined in ticks using nested polymerase chain reaction (nPCR). The acaricidal efficacy of DPP-treated hair was 100% within 1 h of tick release on every study day for both *I. ricinus* and *I. scapularis*. The speed of kill associated with DPP was sufficiently fast to prevent tick attachment and engorgement, and, consequently, to prevent the acquisition of *B. burgdorferi*. In the F-treated group, the acaricidal efficacy observed at 12 h, throughout the study, was <20% and <28% for *I. scapularis* and *I. ricinus*, respectively. Furthermore, tick feces were observed in the FUs, and several female ticks (*I. scapularis* (*n* = 55) and *I. ricinus* (*n* = 94)) tested positive for *B. burgdorferi*. The results provide proof of concept for the use of an ex vivo model based on an artificial feeding system to compare two ectoparasiticides against the acquisition of *B. burgdorferi* by *I. ricinus* and *I. scapularis*. In addition, our results demonstrate the superiority of DPP compared to F in the speed of acaricidal activity against ticks, as well as in preventing the acquisition of *B. burgdorferi*.

## 1. Introduction

Among ectoparasites, ticks are very important and harmful blood-sucking Acari that parasitize a broad range of vertebrates and occasionally bite humans [1,2]. During their blood meal, ticks may transmit a large number of pathogens, including bacteria, viruses, protozoa, and nematodes, that cause disease in humans and their pets or livestock [2]. In addition to tick-borne pathogen transmission, tick bites can cause discomfort, irritation, serious physical damage, dermatitis, a reduction in live weight, and anemia in their hosts and even lead to severe paralysis caused by the injection of a toxin by certain ticks while feeding [3,4].

The prevention of tick-borne diseases (TBDs) in companion animals is commonly achieved through the periodic administration of products that can repel and/or rapidly kill arthropods, thus preventing or interrupting feeding before pathogens are transmitted to the hosts [4]. This prevention strategy also reduces the risk of the transmission of pathogens of zoonotic importance [5]. At present, several ectoparasiticides against ticks are available, but they differ widely in terms of their pharmacokinetics and pharmacodynamics. The efficacy of an ectoparasiticide against a given arthropod vector, such as ticks, does not imply blocking VBP transmission [4]. The reduction in the risk of TBD transmission to the host is related to the speed of transmission of the specific pathogen by its tick vector [6]. By definition, the transmission time is the minimum period required from blood feeding initiation by the tick vectors to actual pathogen transmission to the vertebrate host [4], meaning that the probability of the transmission of VBPs to the host increases when transmission times are short. The occurrence of fast transmission has indeed been reported for some tick-borne viruses, such as Powassan virus and tick-borne encephalitis virus, which are transmissible within 15 and 60 min after tick attachment, respectively [7,8]. Bacterial pathogens such as *Ehrlichia* spp. and *Anaplasma* spp. are transmissible to the host within 3 and 24 h of tick attachment, respectively [9]. According to these considerations, the pharmaceutical products used to prevent VBP transmission should be characterized by a fast onset of killing activity and/or repellency against arthropods [4].

Fluralaner (F, Bravecto Chew, MSD Animal Health Innovation GmbH, Schwabenheim, Germany) is a systemically distributed isooxazoline-class insecticide and acaricide that delivers persistent efficacy against numerous ixodid ticks, including adult *I. ricinus* for 12 weeks following the oral administration of a single commercial dose (25 to 56 mg/kg body weight) in dogs. Its onset of effect is expected within 12 h of attachment for *I. ricinus* [10]. Vectra^®^ 3D (DPP, Ceva Santé Animale SA, Libourne, France) is a topical repellent ectoparasiticide combining three active substances—dinotefuran, pyriproxyfen, and permethrin (DPP)—at 54.00 mg, 4.84 mg, and 397.00 mg per mL, respectively. The product is used for dogs with the minimum recommended dose of 0.12 mL/kg body weight to treat and prevent flea and tick infestations (including adult *I. ricinus* and *I. scapularis*) as well as to repel sandflies, mosquitoes, and stable flies with persistent insecticidal and repellent activity for 1 month [11].

Recently, several artificial feeding systems have been implemented to feed arthropods as potential alternatives to replace the use of experimental animals in some situations, especially in vector colony maintenance and arthropod–pathogen interaction studies. In fact, the use of live animals for this objective is only authorized after bioethical certification by animal care committees under frequently revised protocols. It is advised that protocol evaluations carried out by bioethical committees consider the 3Rs principle (replacement, reduction, refinement) related to animal welfare [12]. Therefore, the scientific community focused on vector–host–pathogen interaction studies needs to strongly consider these artificial feeding options as bioethical alternatives. The effectiveness of DPP in preventing the acquisition of *Borrelia burgdorferi* sensu stricto by *Ixodes ricinus* and *Ixodes scapularis* ticks was assessed using an ex vivo feeding system [13]. Using the same ex vivo model, the current study was conducted to compare the prevention efficacy of these two ectoparasiticides (DPP versus F) against the acquisition of *B. burgdorferi* by adult *I. scapularis* and *I. ricinus*, the most important tick vectors of the agent of human and canine LB. The speed of kill and anti-feeding efficacy of the respective products were also studied.

## 2. Materials and Methods

### 2.1. Animals and Treatment Administration

This study was a parallel-group-designed, randomized, single-center, negative-controlled efficacy study in which thirty-two beagle dogs were included (Figure 1). The dogs were obtained from Clinvet’s colony, healthy, >6 months of age, and between 10 and 15 kg body weight at inclusion, and no dogs were infested with ticks nor treated with any topical or systemic acaricide/insecticide drug within the 3-month period of this study. The study protocol was approved by the Institutional Animal Care and Use (N° CG1079_CVM21).

The dogs were assigned to three groups based on gender and body weight: a fluralaner (F, Bravecto^®^) treatment group (*n* = 8), administered a single oral treatment on day 0 at the recommended dose (one 500 mg Bravecto chewable tablet for dogs weighing > 10 kg to 20 kg); a dinotefuran–permethrin–pyriproxyfen (DPP, Vectra^®^ 3D) treatment group (*n* = 8), topically treated on day 56 at the recommended dose (one applicator tube of 3.6 mL for dogs weighing > 10 kg to 25 kg); and a no-treatment control group (*n* = 16). It is noteworthy that for Vectra^®^ 3D, the three active substances were still detected in dog hair one month after treatment, which explains why the product was administered on day 56 for the DPP-treated group.

The quantity/volume of the investigational veterinary product (IVP) administered to each dog was calculated according to the individual body weight determined on day 7 (F-treated group) and day 49 (DPP-treated group). The fluralaner chewable tablets were administered via placement in the back of the oral cavity over the tongue to initiate swallowing, according to the label instructions. DPP was administered by parting the hair and applying the appropriate volume of DPP directly onto the skin in a continuous line from the base of the tail along the middle of the back to between the shoulder blades, according to the label instructions.

### 2.2. Hair and Blood Sample Collection

Hair and blood samples were collected from each dog on days 58 (day 2 after administration for DPP-treated group), 63 (day 7 after administration for DPP-treated group), 70 (day 14 after administration for DPP-treated group), 77 (day 21 after administration for DPP-treated group), and 84 (day 28 after administration for DPP-treated group), covering the entire efficacy period of each product (Figure 1). Hair (about 3 g/dog) was collected from each individual by brushing the whole body (using single-use brushes for DPP- and F-treated dogs; Figure 2). For the control group, hair was collected from the paired dogs and mixed well to obtain a homogeneous sample, which was used during the membrane preparation. Blood (about 15 mL/dog) was collected aseptically into 1.8 mL sodium citrate tubes according to the following schedule: (a) once every other day for four days from eight dogs in the no-treatment control group and (b) once a day for one or two days (depending on tick survival) from all dogs in the DPP- or F-treated groups.

### 2.3. Ticks

Adult female and male laboratory-reared pathogen-free *I. ricinus* and *I. scapularis* were obtained from Clinvet’s colony initiated in 2012 using adult specimens from Utrecht (The Netherlands) and Georgia (USA), respectively.

The pathogen-free larval and nymphal ticks were maintained through routine passage on specific pathogen-free rabbits. Adult ticks were gorged to repletion on pathogen-free sheep. All tick stages from this laboratory-reared colony were maintained in aired vials placed inside boxes kept at 10–12 °C and a relative humidity (RH) of ~90% (using castor oil or saturated potassium nitrate). To increase their willingness to feed, the ticks were transferred to a 22 °C, 90% RH, and 16:8 h light/dark cycle environment (Panasonic Corporation, Osaka, Japan) at least 5 days prior to the tick feeding challenges.

### 2.4. Bacterial Strain

The *B. burgdorferi* sensu stricto strain B31 (Bbss; ATCC 35210; American Type Culture Collection, Manassas, VA, USA) was used in this experiment. Cultures were grown in complete Barbour–Stoenner–Kelly medium (BSK-H; Sigma, St. Louis, MO, USA) at 34 °C. The spirochete cell viability and concentration were determined by counting spirochetes using dark-field microscopy at 20× and counting the organisms with a Petroff Hausser chamber (Hauser Scientific, Horsham, PA, USA).

### 2.5. Membrane and Tick Feeding Unit Preparation

The feeding unit (FU) design was prepared based on an in-house method described previously [13]. The feeding units were made by stacking two plexiglass tubing chambers onto each other (Figure 2). The two chambers were separated by a previously prepared silicone-based membrane, which was gently fixed to the feeding chamber using mastic silicone glue (SikaSeal^®^, Dublin, Ireland). The feeding membranes consisted of goldbeater’s skin originally made from bovine intestine (Preservation Equipment Ltd., Diss, UK) with a thickness of 30 μm, which were treated with a thin layer of silicone rubber (Smooth-On, Inc., East Texas, PA, USA) mixture to improve the softness, resulting in a final membrane thickness of 100–140 µm. Just after adding the silicone mixture, the dog hair previously collected was added to the membrane to form a thin layer of hair (0.007 g/cm^2^). The membrane was allowed to polymerize overnight at room temperature.

### 2.6. Blood Preparation

The collected blood was supplemented with glucose (Sigma-Aldrich, St. Louis, MO, USA) to a concentration of 2 mg/mL. The blood was then spiked with BSK-H medium containing spirochetes to obtain a final concentration of 10^5^ cells/mL. The blood was changed every 12 h for 3 days, and during this process, all chambers containing blood (chamber 2) were washed three times with distilled water, and at least twice with warm phosphate-buffered saline (PBS) containing gentamicin (10 mg/mL; Sigma-Aldrich) to remove blood residue. The chambers were then left to dry for 5 min under a biosafety cabinet before adding blood.

### 2.7. Tick Seeding, Incubation, Mortality, and Engorgement Assessment

Each FU was filled with 40 adult ticks (1:1 sex ratio) of *I. ricinus* or *I. scapularis* on days 58, 63, 70, 77, and 84. Three milliliters of prepared canine blood containing *Bb*ss was added into each FU and then sealed with sterile parafilm. The FUs were incubated separately according to the groups inside incubators at 37 °C with a humidity >90%. Immediate tick mortality was assessed 1, 2, and 12 h after tick seeding. Tick attachment, engorgement, and feces presence were observed from 1 h up to 72 h after tick seeding. The collected tick specimens were categorized with the naked eye as engorged or non-engorged. However, a stereomicroscope was used to detect traces of a blood meal when differentiation was not possible with the naked eye.

### 2.8. Detecting Ticks Infected with Bbss

After each collection time point, the ticks were washed twice in phosphate-buffered saline (PBS, Sigma-Aldrich) and dried. The ticks were then labeled and stored at −20 °C before being sent to the “Institut Pasteur” for analysis. DNA was purified from whole ticks using the DNeasy^®^ blood and tissue kit protocol from Qiagen according to the manufacturer’s recommendations (Qiagen, Hilden, Germany).

The DNA was individually extracted from all partially engorged females regardless of the group, while in cases where any engorged specimen was noted, three ticks per feeder per time point were randomly selected and the DNA was isolated from individual ticks. The uptake of *Bb*ss organisms was examined using nPCR, amplifying a fragment of 250 base pairs (bp). The primers INS1 [5′ GAAAAGAGGAAACACCTGTT 3′] and S23R [5′ TCGGTAATCTTGGGATCAAT 3′] were added to each reaction mixture at a final concentration of 0.5 μM for the primary amplification [14]. The optimized cycling conditions for the primary amplification involved an initial 4 min denaturation at 94 °C, followed by 35 cycles, each consisting of 1 min of denaturation at 94 °C, 1 min of annealing at 56 °C, and 1 min of extension at 72 °C. These 35 cycles were followed by a 10 min extension at 72 °C, and the reaction products were subsequently maintained at 4 °C and used as a primary template in the nested amplification. The nested amplifications used 3 μL of the primary PCR product as a template in a total volume of 50 μL. The primers RRB [5′ AAGCTCCTAGGCATTCACCATA 3′] and RRC [5′ CTGCGAGTTCGCGGGAGAG 3′] were added at a final concentration of 0.5 μM per reaction for the nested amplification. The nested cycling conditions were as described for the primary amplification, except the annealing step, which was conducted at 60 °C. The reaction products were analyzed using agarose 2% gel electrophoresis after 45 min of DNA migration at 120 Volts. Negative controls were processed with DNA-free water and DNA from non-infected ticks, and the positive control was the genomic DNA of *Bb*ss.

### 2.9. Blinding

All personnel involved with daily general health observations, weighing enrolment, post-treatment observations, hair and blood sample collection, in vitro feeding set up and evaluation and tick analyses were blinded to the treatment groups. Moreover, personnel involved in the assessments differed from personnel involved in sampling for all groups.

### 2.10. Statistical Analysis

Data were recorded using a standardized Data Capture Form (DCF), entered into a Microsoft Excel^®^ (MS Excel 2016) spreadsheet, and analyzed using Prism 7 software (GraphPad Software, Inc., San Diego, CA, USA). Statistical differences were measured using the Chi-square two-tailed test (or with Yates correction when necessary) for the three group comparisons. The target parameters were tick counts at 1, 2, and 12 h post tick seeding as well as positive ticks for *Bb*ss. A difference of *p* < 0.05 was considered significant.

The efficacy was calculated using Abbott’s formula (Abbott 1987):$$Acaricidal\;efficacy\left(\%\right)=100\times \frac{MC-MT}{MC}$$
where MC and MT are the arithmetic mean of live ticks in the control and treated groups, respectively.
$$Antif\text{-}eeding\;efficacy\left(\%\right)=100\times \frac{MC-MT}{MC}$$
where MC and MT are the arithmetic mean of engorged ticks in the control and treated groups, respectively.

The preventive efficacy of DPP against *Bb*ss acquisition by ticks was calculated as a percentage, as follows:$$Prevention\;efficacy\left(\%\right)=100\times \frac{IcC-IcT}{IcC}$$
where IcC and IcT are the number of female ticks positive for *Bb*ss based on PCR in the non-treatment control group and in the treated groups, respectively.

## 3. Results

### 3.1. Animal Health Observations

There were no adverse events related to treatments for either treated group (F and DPP) on any post-treatment assessment day. This is in line with the safety profile of each product.

### 3.2. Mortality Assessment of Ticks and Acaricidal Efficacy at 1, 2, and 12 h

In the DPP-treated group, the proportion of dead ticks (both tick species) assessed 1 h post tick seeding was 100% (AM = 40) at each subsequent challenge. The immediate acaricidal efficacy of DPP-treated hair against *I. ricinus* and *I. scapularis* reached 100% at the 1 h counts (Figure 3 and Figure 4).

In the F-treated group, some ticks were found dead after 2 h of incubation, but this mortality did not exceed 18% (between 3.5 and 17.81%) for *I. scapularis* and 28% (between 0.93 and 27.5%) for *I. ricinus*. The AM of live *I. scapularis* and *I. ricinus* ticks assessed 12 h after tick seeding on days 58, 63, 70, 77, and 84 was 33.5, 32.9, 32.1, 37.5, and 39.6 and 33.5, 35.4, 31.3, 29.0, and 39.6, respectively. The speed of kill efficacy for the F-treated group ranged from 0% (at the 1 h time point on all days for both tick species) to 27.5% (at the 12 h time point on day 77) for *I. ricinus* (Figure 3) and 19.7% (at the 12 h time point on day 70) for *I. scapularis* (Figure 4). Except for day 84, there was a significant difference (*X*^2^ test, *p*-value < 0.0006) in the rate of dead ticks between the F-treated and control groups assessed at 12 h after tick seeding.

### 3.3. Anti-Feeding Efficacy and Feces Observation

The median engorgement rate of the female ticks in the control group was 26.2% (210/800) and 16.25% (130/800) for *I. scapularis* and *I. ricinus*, respectively. The start of tick engorgement was confirmed by the presence of feces inside the FUs (Figure 5).

At each time point challenge, no attachment or presence of feces was observed in the DPP-treated group (Figure 6).

In the DPP-treated group, the anti-feeding effect was 100% against both tick species throughout the study period (Table 1). In the F-treated group, tick attachment, engorgement, and feces presence were observed at each time point. The anti-feeding efficacy (based on AM) 72 h after tick seeding on days 58, 63, 70, 77, and 84, respectively, was 23.1, 98.3, 51.5, 60.9, and 9.1% for *I. scapularis* and 62.5, 47.1, 0, 36.8, and −105.3% for *I. ricinus* (Table 1).

### 3.4. Bbss Acquisition

In the untreated control group, *Bb*ss DNA was detected in a total of 81.4% (153/188) and 92.9% (119/128) of the partially engorged female *I. scapularis* and *I. Ricinus* ticks, respectively (Table 2). All of the randomly selected female ticks from the DPP-treated group (*I. ricinus*: *n* = 120 and *I. scapularis*: *n* = 120) tested negative for *B. burgdorferi* (Table 2). Thus, the preventive effect provided by DPP-treated dog hair against the acquisition of *Bbss* by both tick species was 100% throughout the study period.

In the F-treated group, *Bb*ss DNA was detected in a total of 87% (94/108) and 83.3% (55/66) of the partially engorged female *I. ricinus* and *I. scapularis* ticks, respectively (Table 2). The prevention efficacy determined for F ranged from 14.8 to 55.6% for *I. ricinus* and from 34.1 to 64.1% for *I. scapularis* (Figure 7).

## 4. Discussion

Although veterinary pharmaceutical products have been extensively studied on target animals (e.g., dogs, cats, cattle), much less work has been conducted using ex vivo models. Our study expanded the current knowledge of the usefulness of in vitro tick feeding systems in the evaluation of the effectiveness of a given molecule in killing ticks and even in the prevention of bacterial (e.g., *Bbss*) acquisition by tick vectors. Our results add value to previously published work attempting to use different feeding methods to feed arthropods artificially, since these in ex vivo models not only contribute to the adoption of the 3Rs principle and non-animal testing strategies [15] but also help to reduce the delays and costs associated with product release testing [16]. In addition, these in vitro assays provide researchers with a rapid tool to investigate the transmission of VBDs [17,18] as well as to test the effectiveness of anti-tick vaccines [19] or topical acaricides [13].

Despite experiencing the same experimental conditions (e.g., the application of dog hair on the membrane as a chemical and mechanical stimulus, RH ≥ 90%, and temperature of 37 °C) during tick incubation, the engorgement rates obtained here (8.1% and 13.1% for *I. ricinus* and *I. scapularis*, respectively) was lower than those reported in our previous challenge, in which we obtained median engorgement rates of 27.8 and 39.4% for *I. ricinus* and *I. scapularis*, respectively, after 4 days of tick incubation [13]. We observed that the ticks in this experiment appeared less active than the ones used previously, which may explain the difference in the engorgement rate.

The study was laid out over a period of 3 months (from day 0 to day 84) because it compared two products with different durations of protection, with one licensed for monthly use (DPP) and the other being a 3-month product (F). The challenge period during which the hair and blood samples were collected from dogs to test acaricidal efficacy and *Bbss* prevention was chosen to include the whole claimed efficacy period for DPP and the end of the claimed efficacy period for F. The first tick challenge was carried out on day 58, corresponding to day 2 for the DPP group, and the last tick challenge was performed on day 84, corresponding to day 28 for the DPP group.

DPP demonstrated 100% effectiveness in just 1 h of tick contact with treated hair, and this persisted for one month (days 56, 63, 70, 77, and 84), supporting the pharmacokinetics of DPP, which showed a rapid distribution and maximum concentration shortly (2 days) after application and persisted towards the end of the treatment period, when the three active substances were still detected in different zones of the hair coat [11]. The speed of kill action against *I. ricinus* and *I. scapularis* aligns with the previously obtained results in which 100% of the ticks were dead between 1 and 2 h of contact with treated hair [13]. In addition to its fast acaricidal effects, DPP is known to exert a potent repellent and knockdown effect via contact irritancy against ticks and other host-seeking arthropods, and this is due to its permethrin component [11,20]. This significant repellent property could explain why, in the FUs containing DPP-treated hair, no *I. ricinus* and *I. scapularis* were found attached/engorged, and all randomly tested specimens were negative for *Bbss*.

For the F-treated dogs, a moderate acaricidal efficacy was noted at 2 h after tick seeding, which then increased to reach a maximum of 27.5% on day 77 and 19.7% on day 70 at 12 h against *I. ricinus* and *I. scapularis*, respectively. By comparison, this speed of kill efficacy against *I. ricinus* was lower than that reported in an anterior study in which the authors noted a persistent speed of kill against *I. ricinus* ranging from 33.2% in week 4 to 7.8% in week 12 after treatment for 4 h; from 96.8% in week 4 to 45.8% in week 12 after treatment for 8 h; and from 99.7% in week 4 to 98.3% in week 12 after treatment for 12 h [21]. This could be explained by the mode of action of systemic acaricides like fluralaner, which require some uptake of blood meal by the ticks from the treated animals before being killed. Furthermore, the variability in acaricidal efficacy seen at these early assessment time points (2 and 12 h) depends largely on the actual time point at which the ticks attach and start blood feeding after seeding (ex vivo model) or infestation (animal model). This mode of action differs for topically applied products like DPP that express topical efficacy and act via contact, thus working directly during infestation and prior to potential attachment and feeding [11,22].

The detection of *Bbss* DNA in partially engorged ticks from F-treated dogs was proof of the failure of F to prevent *B. burgdorferi* uptake. In fact, the speed of kill of ticks with this product was not sufficiently fast to prevent blood meals and *B. burgdorferi* acquisition throughout the study period. However, the speed of kill efficacy is the most important indicator of the potential of a given product to block the transmission of disease pathogens. In general, the transmission of VBDs to the host does not usually start immediately after tick attachment but after a certain time period often called the “grace period” [9,23]. The length of the grace period diverges widely between different pathogens, principally depending on their physical location within the tick (midguts, hemolymph, or salivary glands) and whether they need to undergo physiological, phenotypical, or life-stage changes prior to becoming infectious for susceptible vertebrate hosts [23]. For example, the inoculation of an infectious dose of *B. burgdorferi* through the tick bite is generally considered to occur between 16 and 48 h of tick attachment [9]. However, it has been observed that for certain *Borrelia* strains (e.g., *Bbss* strain BRE 13) and for certain tick species (e.g., *I. ricinus*), the transmission may occur within the first 12 h [14]. This experimental observation was reinforced by the detection of *B. burgdorferi* DNA in the salivary glands of unfed questing *Ixodes* spp. ticks [24]. Therefore, in the light of the present study results and the risk reduction for the transmission of disease pathogens to pets, the most effective preventive measures should rely on the prevention of tick attachment and blood feeding.

## 5. Conclusions

Hair collected from DPP-treated dogs provided a 1 h acaricidal efficacy of 100% against adult *I. ricinus* and *I. scapularis* ticks throughout the study. The speed of kill of DPP against *Ixodes* spp. was sufficient to prevent the acquisition of *B. burgdorferi*. In contrast, the 12 h acaricidal efficacy of the systemic compound “Bravecto” against the same tick species was not enough to prevent tick engorgement and *B. burgdorferi* acquisition. This study was performed without exposing the dogs to the vectors or to the pathogen.

## Figures and Tables

**Figure 1 microorganisms-12-00202-f001:**
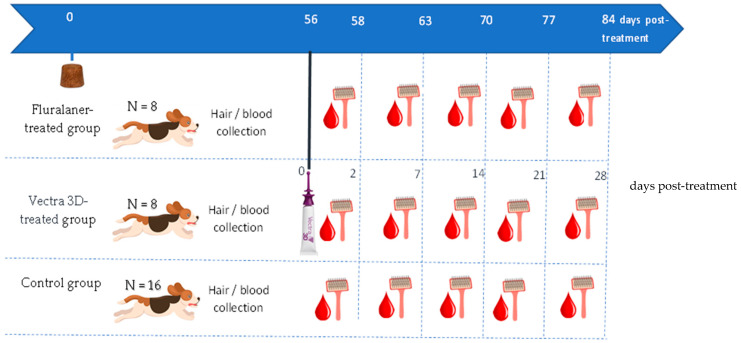
Schematic representation of treatment and blood/hair sampling time point and number of animals per group.

**Figure 2 microorganisms-12-00202-f002:**
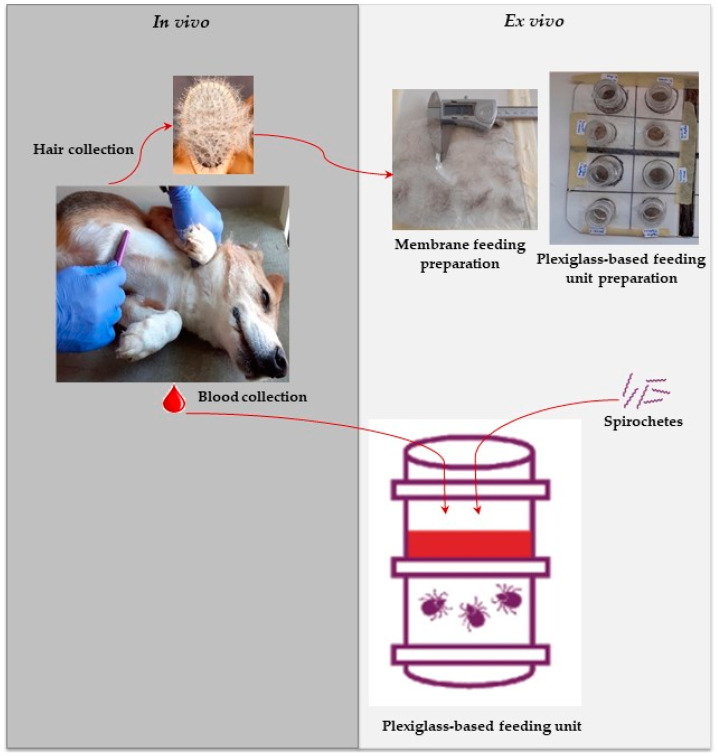
Study design. Schematic view of the experiments performed on dogs (in vivo) and in ex vivo.

**Figure 3 microorganisms-12-00202-f003:**
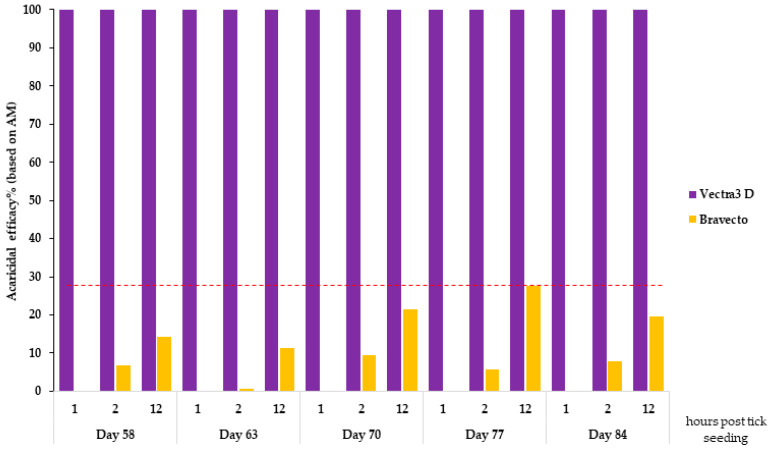
Acaricidal efficacy against *I. ricinus* 1, 2, and 12 h after tick seeding.

**Figure 4 microorganisms-12-00202-f004:**
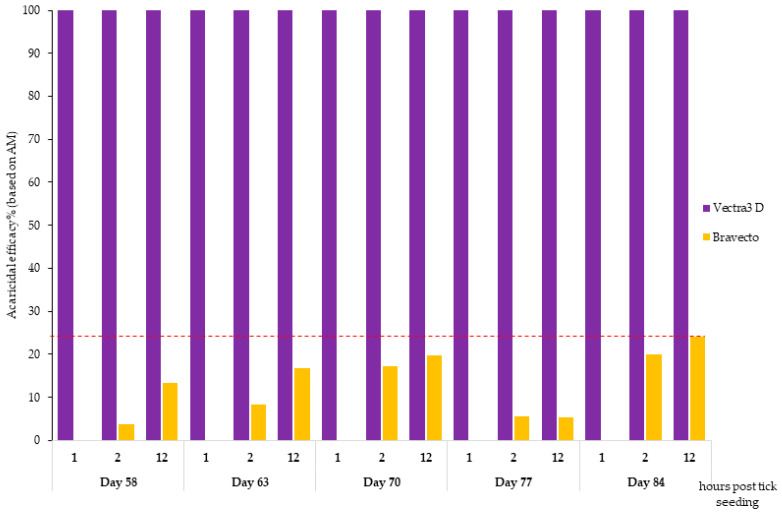
Acaricidal efficacy against *I. scapularis* 1, 2, and 12 h after tick seeding.

**Figure 5 microorganisms-12-00202-f005:**
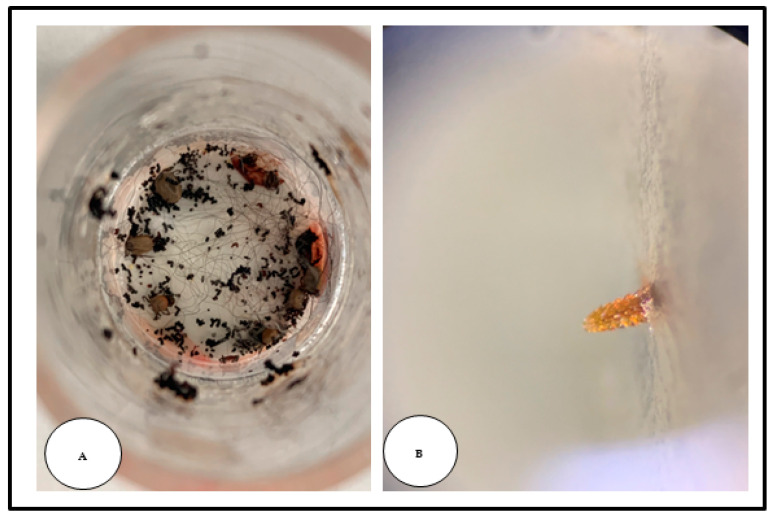
Adult *Ixodes* ticks during artificial blood feeding. (**A**) *Ixodes scapularis* females attached to the silicone-based membrane with a dog’s hair (no-treatment control group). Abundant feces are apparent around the ticks, which demonstrated their active feeding. (**B**) Outside view of the adult tick hypostomes perforating the silicone membrane.

**Figure 6 microorganisms-12-00202-f006:**
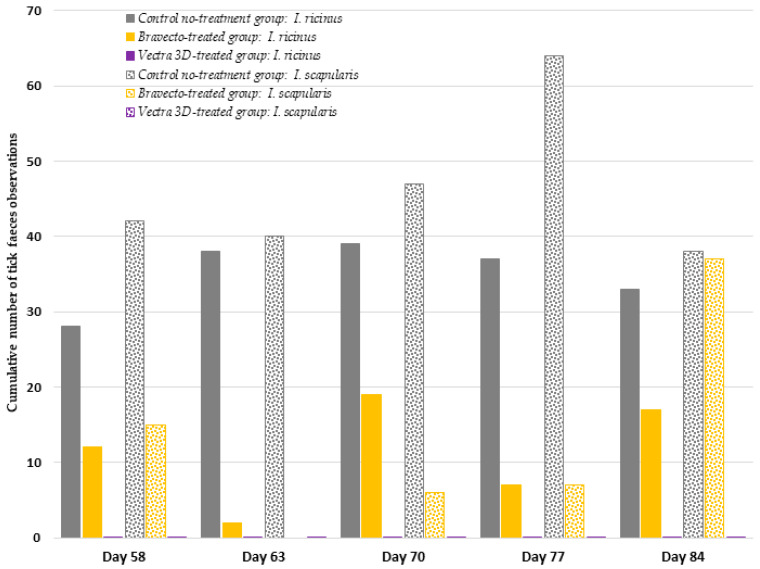
Cumulative number of tick feces observations noted in feeding units at each time point.

**Figure 7 microorganisms-12-00202-f007:**
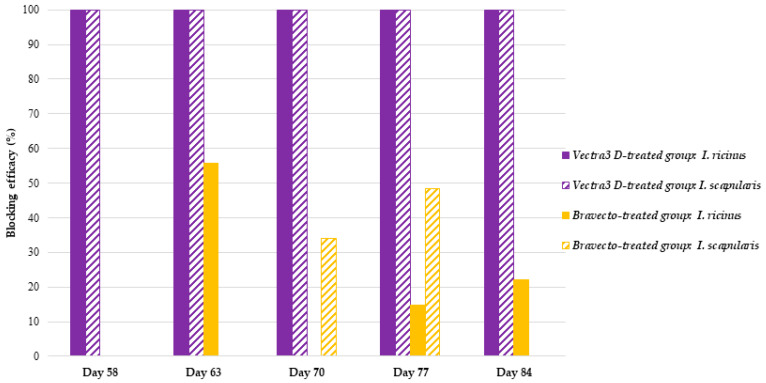
Preventive effect provided by Vectra 3D- or Bravecto-treated dog hair against the acquisition of *B. burgdorferi* by *I. ricinus* and *I. scapularis* ticks at each time point.

**Table 1 microorganisms-12-00202-t001:** Anti-feeding efficacy (based on arithmetic means) of DPP- and fluralaner-treated dogs against *Ixodes ricin*us and *Ixodes scapularis* ticks.

Days	Control (AM)		Bravecto (AM)		Anti-Feeding Efficacy (%)		Vectra 3D (AM)		Anti-Feeding Efficacy (%)	
	*Ixodes ricinus*	*Ixodes scapularis*	*Ixodes ricinus*	*Ixodes scapularis*	*Ixodes ricinus*	*Ixodes scapularis*	*Ixodes ricinus*	*Ixodes scapularis*	*Ixodes ricinus*	*Ixodes scapularis*
58	1	3.3	0.4	2.5	62.5	23.1	0	0	100	100
63	2.1	7.4	1.1	0.1	47.1	98.3	0	0	100	100
70	6	8.5	6	4.1	0	51.5	0	0	100	100
77	4.8	5.8	3	2.3	36.8	60.9	0	0	100	100
84	2.4	1.4	4.9	1.3	/	9.1	0	0	100	100

Only females were considered during the assessment. Each feeding unit contained 20 female ticks. AM: arithmetic mean.

**Table 2 microorganisms-12-00202-t002:** *Borrelia burgdorferi* DNA detection in female ticks at each time point.

Days	Control No-Treatment Group Positive/Total Tested (%)		Bravecto-Treated Group Positive/Total Tested (%)		Vectra 3D-Treated Group Positive/Total Tested (%)	
	*Ixodes ricinus*	*Ixodes scapularis*	*Ixodes ricinus*	*Ixodes scapularis*	*Ixodes ricinus*	*Ixodes scapularis*
58	8/8	21/26 (80.7)	NA	NA	0/24 (0)	0/24 (0)
63	18/18 (100)	50/58 (86.2)	8/9 (88.8)	0/1 (0)	0/24 (0)	0/24 (0)
70	48/48 (100)	41/49 (83.7)	49/58 (84.4)	27/33 (81.8)	0/24 (0)	0/24 (0)
77	27/35 (77.14)	31/44 (70.4)	23/25 (92)	16/20 (8)	0/24 (0)	0/24 (0)
84	18/19 (94.9)	10/11 (90.9)	14/16 (87.5)	12/12 (100)	0/24 (0)	0/24 (0)
Total	119/128 (92.9)	153/188 (81.4)	94/108 (87)	55/66 (83.3)	0/120 (0)	0/120 (0)

## Data Availability

The data presented in this study are available in the article.

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
