# Peer review of "Comparative Evaluation of the Efficacy of Two Ectoparasiticides in Preventing the Acquisition of Borrelia burgdorferi by Ixodes scapularis and Ixodes ricinus: A Canine Ex Vivo Model"

_microorganisms, 2024, doi:10.3390/microorganisms12010202_

Round 1

Reviewer 1 Report

Comments and Suggestions for Authors

1. The paper describes comparative ex vivo evaluation of two different acaricides for prevention of acquisition of Borrelia burgdorferi by two species of vector ticks.  The authors claim the results provide proof-of-concept for use of an ex vivo artificial feeding model for comparing two ectoparasiticides in relation to acquisition of the Lyme borreliosis spirochaete by feeding ticks.  The ex vivo model has previously been reported – the new information is a comparison of two different acaricides.  This comparison appears confounded by the fact the two acaricides operate in different ways – one is topical applied directly to the hair along the back midline of dogs whereas the other is systemic, orally delivered.  

1a. Hair was collected from each dog on days 58, 63, 70, 77 and 84.  This is mis-leading in Figs. 2, 3, 6, Table 1 and Table 2 as these days refer to the duration after oral administration of Fluralaner.  For DPP, the times represent 2, 7, 14, 21 and 28 after topical administration of the repellent ectoparasiticide.  DPP is licensed for monthly use; the observed results (100% lethality at all time points) are similar to those reported in a previous publication by the authors. 

1b. Fluralaner is licensed as a 3-month product.  The challenge period during which hair and blood samples were taken included the whole claimed efficacy period for DPP and the end of the claimed efficacy period for Fluralaner.  Clearly the comparison was not strictly comparable and was biased towards DPP treatment.

1c.  Ticks were exposed simultaneously to blood and hair from the acaricide-treated dogs (and controls).  Given that DPP is a repellent applied to the hair of dogs whereas Fluralaner is orally administered, the design again appears biased in favour of DPP treatment.  For example, there was no comparison of tick feeding on blood from the acaricide treated groups but with hair from the control group.

1d.  Hair was collected by brushing the whole body of dogs.  On dogs, ticks are often found on the head around the eyes, on ears, between toes, and on the tail and groin.  What evidence is there that the hair samples reflect the environment where ticks are commonly found on dogs?

2.  All personnel involved in data collection were blinded to treatment groups.  Presumably, personnel were not blinded to the control group as sample collection was different for controls compared with acaricide-treated dogs.

3. As discussed in the paper, the most important criterion for an effective acaricide is to protect against the transmission of pathogens by ticks to dogs and other animals they feed on (including humans), i.e. to protect against pathogen transmission from ticks.  The ex vivo model described in the paper evaluates the ability of acaricides to prevent transmission to ticks, not from ticks.  The value of the model therefore appears limited.  

Line 234-235:  This sentence does not make sense.

Line 254-255:  The results for DPP treatment are not shown in Fig. 5.

Comments on the Quality of English Language

The English requires some editing.

Author Response

1- The paper describes comparative ex vivo evaluation of two different acaricides for prevention of acquisition of Borrelia burgdorferi by two species of vector ticks.  The authors claim the results provide proof-of-concept for use of an ex vivo artificial feeding model for comparing two ectoparasiticides in relation to acquisition of the Lyme borreliosis spirochaete by feeding ticks.  The ex vivo model has previously been reported – the new information is a comparison of two different acaricides.  This comparison appears confounded by the fact the two acaricides operate in different ways – one is topical applied directly to the hair along the back midline of dogs whereas the other is systemic, orally delivered.  

1a. Hair was collected from each dog on days 58, 63, 70, 77 and 84.  This is mis-leading in Figs. 2, 3, 6, Table 1 and Table 2 as these days refer to the duration after oral administration of Fluralaner.  For DPP, the times represent 2, 7, 14, 21 and 28 after topical administration of the repellent ectoparasiticide.  DPP is licensed for monthly use; the observed results (100% lethality at all time points) are similar to those reported in a previous publication by the authors. 

1b. Fluralaner is licensed as a 3-month product.  The challenge period during which hair and blood samples were taken included the whole claimed efficacy period for DPP and the end of the claimed efficacy period for Fluralaner.  Clearly the comparison was not strictly comparable and was biased towards DPP treatment.

Answer: The authors thank the reviewer for all these pertinent remarks/suggestions. All of them have been taken into consideration through the manuscript.

As for the first remark/question “this comparison appears confounded by the fact the two acaricides operate in different ways – one is topical applied directly to the hair along the back midline of dogs whereas the other is systemic, orally delivered” and “1b. Fluralaner is licensed as a 3-month product.  The challenge period during which hair and blood samples were taken included the whole claimed efficacy period for DPP and the end of the claimed efficacy period for Fluralaner.  Clearly the comparison was not strictly comparable and was biased towards DPP treatment”, the authors believe that this type of comparison is justified for several reasons:

  • Both products (Vectra 3D and Bravecto) have their marketing authorizations, so the pharmacokinetics and pharmacodynamic are known for each investigated product.
  • Bravecto is licensed for quarterly use, whereas Vectra 3D is licensed for monthly use, and this is related to the persistent (residual) efficacy approved against ixodid ticks (and other bloodsucking arthropods) on target animal (dogs). The challenge period during which hair and blood samples were collected from dogs to test the acaricidal efficacy and Borrelia prevention was chosen to include the claimed efficacy period for both products: Vectra 3D (one month) and Bravecto (three months). The first tick challenge was done on day 58 corresponding to day 2 for DPP-treated group, and the last tick challenge was performed on day 84 corresponding to day 28 for DPP-treated group. As we can see, all assessed efficacies have been measured at different time points which are all included in the activity range of each IVP. If the time points 77 and 84 represent the end of the claimed efficacy period for Fluralaner, the time points 21 and 28 also represent the end of the claimed efficacy period for DPP. All these information have been reported/discussed objectively in both “M&M” and “Discussion” sections. An optional study design would have been to apply DPP for 3 consecutive months to mirror the efficacy duration claimed for F. However, this was not justified from a 3R standpoint as the focus of the comparison was at the end of the duration of efficacy for both products. We decided to maintain the weekly assessments over this period as recommended for monthly products (EMA guideline).
  • In this study, the two products have been administrated to dogs at the recommended dose according to the manufacturer's instructions (M&M section).
  • In numerous published data, efficacy comparisons between oral systemic ectoparasiticides and topical repellent ectoparasiticides were performed on animal model (e.g., Jongejan, F., et al. Comparative efficacy of oral administrated afoxolaner (NexGard™) and fluralaner (Bravecto™) with topically applied permethrin/imidacloprid (Advantix®) against transmission of Ehrlichia canis by infected Rhipicephalus sanguineus ticks to dogs. Parasites Vectors 9, 348 (2016). https://doi.org/10.1186/s13071-016-1636-9; Varloud, J. Liebenberg, J.J. Fourie, Comparative preventive efficacy of oral systemic vs. topical repellent ectoparasiticides against early Babesia canis transmission in dogs within 8h of infestation by pre-fed male Dermacentor reticulatus, International Journal of Infectious Diseases 73, 390 (2018). https://doi.org/10.1016/j.ijid.2018.04.4297; Rohdich, N., et al. A randomized, blinded, controlled and multi-centered field study comparing the efficacy and safety of Bravecto™ (fluralaner) against Frontline™ (fipronil) in flea- and tick-infested dogs. Parasites Vectors 7, 83 (2014). https://doi.org/10.1186/1756-3305-7-83). This is a proof of concept of the comparative eligibility of different products.

1c.  Ticks were exposed simultaneously to blood and hair from the acaricide-treated dogs (and controls).  Given that DPP is a repellent applied to the hair of dogs whereas Fluralaner is orally administered, the design again appears biased in favour of DPP treatment.  For example, there was no comparison of tick feeding on blood from the acaricide treated groups but with hair from the control group.

The authors confirm that ticks were exposed to blood and hair collected from each dog at several time points according to the corresponding group (3 different groups) as described in M&M section. We do not understand why the reviewer suggested taking blood from treated dogs but using hair collected from control non-treated group. It seems to us that this makes no sense, since we did not test unknown products. The pharmacokinetics and pharmacodynamics of each IVP are studied/available. So, during the experimental design, authors have taken all these points into account.

1d.  Hair was collected by brushing the whole body of dogs.  On dogs, ticks are often found on the head around the eyes, on ears, between toes, and on the tail and groin. What evidence is there that the hair samples reflect the environment where ticks are commonly found on dogs?

Authors agree with the reviewer about this point. In nature, attachment of ticks at feeding sites on the host also depends on an appropriate array of chemical and physical stimuli, where hair likely plays a key role. We are aware that no “ex vivo” model can replace the “in vivo” model. For artificial feeding systems, we think that we are in the development and optimization steps. Actually, there is no validated model to assess the efficacy of parasiticides in the prevention of VBD.

Obviously, in vivo procedures are stressful and painful for animals due to often massive tick bite burden as well as infection if the ticks are infected. The goal of the 3R principles is to always replace animal experiments whenever possible, reduce the number of animal experiments to the lowest possible, and ensure that the distress inflicted upon the animals is as low as possible. So, we believe that our experimental design is in line with this perspective since than about 80% of the activities previously performed on the dog, such as sedation, tick infestation and tick counts, were avoided.

  1. All personnel involved in data collection were blinded to treatment groups.  Presumably, personnel were not blinded to the control group as sample collection was different for controls compared with acaricide-treated dogs.

This is true but the personnel involved in sampling was also different from the personnel involved in the assessments. We propose to add this sentence to clarify.

“Moreover, personnel involved in the assessments differed from personnel involved in sampling for all groups”; lines 258-259.

  1. As discussed in the paper, the most important criterion for an effective acaricide is to protect against the transmission of pathogens by ticks to dogs and other animals they feed on (including humans), i.e. to protect against pathogen transmission from ticks.  The ex vivo model described in the paper evaluates the ability of acaricides to prevent transmission to ticks, not from ticks.  The value of the model therefore appears limited.  

The authors have not said otherwise. While not tested here, we might expect similar results if the experiment had used infected ticks and uninfected blood, since the transmission prevent effect of each IVP is obtained through its anti-feeding and/or acaricidal action on the ticks. In addition, in anterior study, we demonstrated that DPP prevented Bbss transmission by naturally infected I. scapularis ticks using artificial feeding (DOI:10.13140/RG.2.2.22069.60641).

Line 234-235:  This sentence does not make sense.

The sentence has been deleted.

Line 254-255:  The results for DPP treatment are not shown in Fig. 5.

There were no faeces observed/recorded between in the DPP-group. 

Reviewer 2 Report

Comments and Suggestions for Authors

In the title change the word “blocking”, because they are not blocking, they are avoiding or controlling the transmission of pathogen.

In the summary, also change the term blocking

In the introduction, include information on the importance of using ex vivo models in research to reduce and even eliminate the use of animals in evaluation trials. Adding this information would highlight the importance of the research work reported in this report.

Lines 99-101 Clarify why was the group treated with DPP treated topically on day 56?

Figure 1 maintain the same order in which the experimental groups are described in lines 199-101

 The subtitles 2.5. Membrane and tick feeding unit preparation and 2.6. Membrane and tick feeding unit preparation are the same and do not describe the same, it is suggested to modify them.

In the experimental design, explain why these exposure and treatment times of the evaluated products were used.

In the methodology, they mention that they perform PCR to demonstrate the presence of pathogens, this is important; but at some point did they determine if the pathogens were infectious?

In the discussion, the authors provide information that should be mentioned in the methodology, example lines 304-314 would facilitate a better understanding of the experimental design.

Author Response

The authors thank the reviewer for all these pertinent remarks/suggestions. All of them have been taken into consideration through the manuscript.

  • In the title change the word “blocking”, because they are not blocking, they are avoiding or controlling the transmission of pathogen.

Thank you for this remark. The word “blocking” has been replaced by “preventing”.

  • In the summary, also change the term blocking

Done, line 41.

  • In the introduction, include information on the importance of using ex vivo models in research to reduce and even eliminate the use of animals in evaluation trials. Adding this information would highlight the importance of the research work reported in this report.

The authors agree with the reviewer about this remark, so a paragraph has been added to the introduction as follows: “Recently, several artificial feeding systems have been implemented to feed arthropods as potential alternatives to replace the use of experimental animals in some situations, especially in vector colony maintenance and arthropod–pathogen interaction studying. In fact, the use of live animals for this objective is only authorized after bioethical certification by animal care committees under frequently revised protocols. It is advised that protocol evaluations by bioethical committees consider the 3Rs principle (replacement, reduction, refinement) related to animal welfare [3,4]. Therefore, the scientific community focused on vector-host-pathogen interactions studies needs to strongly consider these artificial feeding options as a bioethical alternative”; lines 85-94.

  • Lines 99-101 Clarify why was the group treated with DPP treated topically on day 56?

The sentence has been edited to include this information: “Noteworthy that for Vectra®3D, the three active substances still detected in dog hair one month after treatment explaining why for DPP-treated group, the product was administered on Day 56”; lines 113-115. Furthermore, in the dissuasion part (lines 314-321), the authors have justified the choice of treatment periods for each product.

  • Figure 1 maintain the same order in which the experimental groups are described in lines 199-101.

The sentence describing the experimental design (lines 199-101) has been reformulated to make it more logical in terms of treatment administration chronology (lines 108-112), so no changes have been made to the figure 1.

  • The subtitles 2.5. Membrane and tick feeding unit preparation and 2.6. Membrane and tick feeding unit preparation are the same and do not describe the same, it is suggested to modify them.

Thank you for this remark. The subtitle 2.6 is now correct.

  • In the experimental design, explain why these exposure and treatment times of the evaluated products were used.

A paragraph has been added to the M&M section to describe the experimental design with more details as recommended by the reviewer: “Hair and blood samples were collected from each dog on days 58 (day 2 for DPP-treated group), 63 (day 7 for DPP-treated group), 70 (day 14 for DPP-treated group), 77 (day 21 for DPP-treated group) and 84 (day 28 for DPP-treated group) covering the entire efficacy period of each product.” (lines 131-135).

In addition, the justification the experimental design including why these exposure and treatment timepoints was discussed in “Discussion section” (lines 366-373).

  • In the methodology, they mention that they perform PCR to demonstrate the presence of pathogens, this is important; but at some point did they determine if the pathogens were infectious?

The authors fully agree with this remark about the importance of checking whether ticks that have acquired Borrelia through artificial feeding are infectious or not. In the present study, we did not check this parameter as we used adult ticks. The best way would be to use nymphs (or larvae) and check after molting whether those ticks still harboring Borrelia alive as well as check whether they are able of transmitting the spirochetes to the host (induce infection). Somme previous studies have demonstrated the adult I. scapularis successfully transmitted infectious B. burgdorferi to animals, when nymphs have been fed by artificial feeding using blood containing spirochetes (https://doi.org/10.1603/0022-2585-38.2.167, https://doi.org/10.1007/s00430-003-0178-x, https://doi.org/10.1038/s41598-018-20208-4).

  • In the discussion, the authors provide information that should be mentioned in the methodology, example lines 304-314 would facilitate a better understanding of the experimental design.

This remark was taken into consideration according to the previously suggested comments. The authors have added more information in the experimental design (Fig. 2 and lines 115-118 and 131-134) and kept this paragraph in the “Discussion section” to avoid too much text in the “M&M section”. 

Reviewer 3 Report

Comments and Suggestions for Authors

I  could not find the reason why so much time elapsed between giving flurlaner and starting the permethrin.

I realize that you have shown the tick/hair/blood methodology elsewhere but it would help to have a similar diagram in this paper since in text it is hard to digest what is happening.

Author Response

The authors thank the reviewer for all these pertinent remarks/suggestions. All of them have been taken into consideration through the manuscript.

  • I  could not find the reason why so much time elapsed between giving flurlaner and starting the permethrin.

The justification the experimental design including why these exposure and treatment timepoints was discussed in “Discussion section”: “The study lay out over a period of three months (from day 0 to day 84) because it compared two products with different duration of protection; one licensed for monthly use (DPP) and the other being a 3-month product (F). The challenge period during which hair and blood samples were collected from dogs to test the acaricidal efficacy and Bbss blocking was chosen to include the whole claimed efficacy period for DPP and the end of the claimed efficacy period for F. The first tick challenge was done on day 58 corresponding to day 2 for DPP group, and the last tick challenge was performed on day 84 corresponding to day 28 for DPP group » (lines 131-135).

In addition, a paragraph has been added to the M&M section to describe the experimental design with more details as recommended by the reviewers (lines 116-118).

  • I realize that you have shown the tick/hair/blood methodology elsewhere but it would help to have a similar diagram in this paper since in text it is hard to digest what is happening.

Thank you for this important remark. A diagram (Fig. 2; lines 193-228) summarizing a part of M&M (membrane feeding and chamber unit preparation, preparation of blood seeded with spirochetes, tick seeding) has been added as suggested.

Reviewer 4 Report

Comments and Suggestions for Authors

Manuscript 'Comparative Evaluation of the Efficacy of two Ectoparasiticides in Blocking the Acquisition of Borrelia burgdorferi by Ixodes scapularis and Ixodes ricinus: A Canine Ex vivo Model" presents important results of the action of different ectoparasiticides on the feeding and acquisition of Borrelia burgdorferi by Ixodes scapularis and Ixodes ricinus. The manuscript is well written and I recommend publication with minimal corrections.

Line 48: "ticks may transmit a large number of vector-borne pathogens (VBPs)". Replace by " ticks may transmit a large number of pathogens"

line 54: "The prevention of VBP transmission in" Reviewer: replace by "The prevention of Tick-Borne Diseases in". Please try to focus on the ticks. See also line 61.

line 97: The study protocol was 97 approved by the Institutional Animal Care and Use. Reviewer: Do you have a protocol number?

Line 117: Reviewer: Replace "specimens" by "samples"

Figures 5 and 6, species names in italics.

Author Response

Manuscript 'Comparative Evaluation of the Efficacy of two Ectoparasiticides in Blocking the Acquisition of Borrelia burgdorferi by Ixodes scapularis and Ixodes ricinus: A Canine Ex vivo Model" presents important results of the action of different ectoparasiticides on the feeding and acquisition of Borrelia burgdorferi by Ixodes scapularis and Ixodes ricinus. The manuscript is well written and I recommend publication with minimal corrections.

The authors thank the reviewer for all these pertinent remarks/corrections. All of them have been taken into consideration through the manuscript.

  • Line 48: "ticks may transmit a large number of vector-borne pathogens (VBPs)". Replace by " ticks may transmit a large number of pathogens".
  • It is done, line 48.
  • line 54: "The prevention of VBP transmission in" Reviewer: replace by "The prevention of Tick-Borne Diseases in". Please try to focus on the ticks. See also line 61.
  • It is done, lines 54 and 61.
  • line 97: The study protocol was 97 approved by the Institutional Animal Care and Use. Reviewer: Do you have a protocol number?
  • The protocol number (N° CG1079_CVM21) has been added, line 108.
  • Line 117: Reviewer: Replace "specimens" by "samples"
  • It is done, line 133.
  • Figures 5 and 6, species names in italics.
  • It is done.

Reviewer 5 Report

Comments and Suggestions for Authors

The manuscript Comparative Evaluation of the Efficacy of two Ectoparasiticides in Blocking the Acquisition of Borrelia burgdorferi by Ixodes scapularis and Ixodes ricinus: A Canine Ex vivo Model is a partial extension of the previous study by the authors performed on Vectra 3D. In previous study the ectoparasiticide was tested on four animals and in this study on eight animals. Aditional ectoparasiticide was tested in the present study and results were compared. Both studies are using the same ex vivo model for artificial feeding of ticks which is a convinient contribution to the 3Rs (reduction, refinement and replacement) principles and non-animal testing strategies. The study could be of interest for the audience of the journal and the majority of the paper is written in sufficient manner. My main concern is the interpretation of the results of molecular detection of B. burgdorferi s.s. in ticks included in the Vectra 3D treated animals. Since the product showed very rapid 100% acaridal effect and ticks didn't feed at all, the molecular testing for B.b.s.s. presence and concludion that ectoparasiticide prevented the transmission of the pathogen is superfluous and is leading to overinterpretation of results. This missleading conclusion is incorporated in the title, since the blocking the acquisition of Borrelia burgdorferi could be analysed only for the Bravecto treated group. Thus, I suggest changing the title to reflect the real results presented in the study and ommit the part of the results, disscussion and conclusion concerning the molecular testing the ticks from Vectra 3D study group. 

Author Response

The manuscript Comparative Evaluation of the Efficacy of two Ectoparasiticides in Blocking the Acquisition of Borrelia burgdorferi by Ixodes scapularis and Ixodes ricinus: A Canine Ex vivo Model is a partial extension of the previous study by the authors performed on Vectra 3D. In previous study the ectoparasiticide was tested on four animals and in this study on eight animals. Aditional ectoparasiticide was tested in the present study and results were compared. Both studies are using the same ex vivo model for artificial feeding of ticks which is a convinient contribution to the 3Rs (reduction, refinement and replacement) principles and non-animal testing strategies. The study could be of interest for the audience of the journal and the majority of the paper is written in sufficient manner.

My main concern is the interpretation of the results of molecular detection of B. burgdorferi s.s. in ticks included in the Vectra 3D treated animals. Since the product showed very rapid 100% acaridal effect and ticks didn't feed at all, the molecular testing for B.b.s.s. presence and conclusion that ectoparasiticide prevented the transmission of the pathogen is superfluous and is leading to overinterpretation of results. This missleading conclusion is incorporated in the title, since the blocking the acquisition of Borrelia burgdorferi could be analysed only for the Bravecto treated group. Thus, I suggest changing the title to reflect the real results presented in the study and ommit the part of the results, disscussion and conclusion concerning the molecular testing the ticks from Vectra 3D study group. 

The authors thank the reviewer for all these pertinent remarks/suggestions. All of them have been taken into consideration through the manuscript. We agree with the reviewer on the need for caution in interpretation the PCR results obtained on Vectra 3D-treated group. It is true that for this group, preventing tick attachment and engorgement is closely related to the fast acaricidal effect against ticks observed during the first hours of exposure to Vectra 3D-treated hairs. We think that the word “blocking” used here is not correct and perhaps “prevention” would be a better word to be used. So, the word “blocking” has been replaced by “preventing” in the title as well as throughout the manuscript.

Specialists in the field of ticks and animal TBDs say that: “The prevention of tick-borne diseases in companion animals is commonly achieved through the periodic administration of products that can repel and/or rapidly kill arthropods, thus preventing or interrupting feeding before pathogens are transmitted to the hosts” (Dantas-Torres, F  et al. 2016; Otranto, D. et al. 2021). This means that when a product X can prevent the blood meal of an arthropod vector, this product would be considered effective in preventing transmission of the infection.

We believe that the molecular testing and conclusion about the prevention of the pathogen uptake are not superfluous. Indeed, tick probing through the membrane (with no or limited feeding) could happen despite the regular observation and without clear visible enlargement of the tick scutum. This was already observed on dogs as ticks were shown to attach, detach and move during the first hours of host infestation (Little et al., 2007; https://pubmed.ncbi.nlm.nih.gov/17904292/). This justifies the tick testing for Borrelia burgorferi in all groups. Moreover, the blocking of the transmission is the finality of the product and can be achieved through different means: either rapid action (if systemic mode of action) or repellent effect (if contact-based mode of action). This is why we selected this criterion as the primary efficacy. The other criteria (anti-feeding, killing effect) were measured to further explain how the primary efficacy was achieved.

Furthermore, during the design/set up of the present experiment, wherever possible, the guidelines for evaluating the efficacy of parasiticides in reducing vector-borne pathogens transmission, recently published by the World Association for the Advancement of Veterinary Parasitology (W.A.A.V.P.) were properly followed. As for artificial feeding systems, we think that we are in the development and optimization steps. Actually, there is no validated model to evaluate the efficacy of parasiticides (including ecto-parasiticides) in the prevention of VBD.

Prevention efficacy calculated for each product was done according to the recommended formula (WAAVP):

, where IcC and IcT the number of positive female ticks for Borrelia based on PCR in the non-treatment control group and in the treated groups, respectively.
